# Enabling long-cycling aqueous sodium-ion batteries via Mn dissolution inhibition using sodium ferrocyanide electrolyte additive

Zhaoheng Liang [1], Fei Tian [1], Gongzheng Yang [1] ✉ & Chengxin Wang [1,2] ✉

Aqueous sodium-ion batteries (AIBs) are promising candidates for large-scale energy storage due to their safe operational properties and low cost. However, AIBs have low specific energy (i.e., <80 Wh kg$^{-1}$) and limited lifespans (e.g., hundreds of cycles). Mn-Fe Prussian blue analogues are considered ideal positive electrode materials for AIBs, but they show rapid capacity decay due to Jahn-Teller distortions. To circumvent these issues, here, we propose a cation-trapping method that involves the introduction of sodium ferrocyanide ($Na_4Fe(CN)_6$) as a supporting salt in a highly concentrated $NaClO_4$-based aqueous electrolyte solution to fill the surface Mn vacancies formed in Fe-substituted Prussian blue $Na_{1.58}Fe_{0.07}Mn_{0.97}Fe(CN)_6 \cdot 2.65H_2O$ (NaFeMnF) positive electrode materials during cycling. When the engineered aqueous electrolyte solution and the NaFeMnF-based positive electrode are tested in combination with a 3, 4, 9, 10-perylenetetracarboxylic diimide-based negative electrode in a coin cell configuration, a specific energy of 94 Wh kg$^{-1}$ at 0.5 A g$^{-1}$ (specific energy based on the active material mass of both electrodes) and a specific discharge capacity retention of 73.4% after 15000 cycles at 2 A g$^{-1}$ are achieved.

Large-scale energy storage systems are essential for the integration of intermittent renewable energies, such as wind, solar and tidal power[1,2]. Benefiting from their nonflammability and abundant resources, aqueous sodium-ion batteries (ASIBs) are regarded as promising candidates for grid energy storage[3,4]. However, the narrow electrochemical stability window of aqueous electrolytes and the material dissolution caused by the high activity of water have restricted the specific energy and cycle stability of these batteries[5]. It remains a substantial challenge to develop compatible electrodes and electrolytes capable of delivering adequate electrochemical energy storage performance. One of the biggest obstacles is the lack of a suitable cathode material that can maintain good structural integrity upon repeated and rapid Na$^+$ (de) insertion.

Various active positive electrode materials have been studied, primarily transition-metal oxides[6,7], polyanionic compounds[8,9], and Prussian blue analogs (PBAs)[10,11]. Among them, PBAs have received increasing interest for their ease of synthesis and easily adjustable properties[12–14]. Recently, some progress has been made in the development of PBAs[15,16], for example, for applications in transparent battery devices[17]. In particular, much attention has been paid to the Mn-based PBAs (MnPB) because of the high working potential of 3.5 V (vs Na$^+$/Na) and environmental friendliness. Unfortunately, MnPB shows poor cycle stability due to the irreversible phase changes arising from Jahn-Teller (JT) distortion[18,19]. The large volume changes (>10%) that occur during the phase transitions continuously trigger surface defects and subsequently lead to internal structural distortions, eventually resulting in the loss of electrochemical activity of the positive electrode due to Mn dissolution. Thus, it is necessary to mitigate or inhibit the JT effect to address this severe challenge.

Previous efforts to suppress Mn dissolution have mainly focused on partial atom doping/substitution in the active positive electrode materials[20,21] or electrolyte optimization[22,23], but none of these

[1]School of Materials Science and Engineering, Sun Yat-sen University, 510275 Guangzhou, P. R. China. [2]State Key Laboratory of Optoelectronic Materials and Technologies, Sun Yat-sen University, 510275 Guangzhou, P. R. China. ✉e-mail: yanggzh5@mail.sysu.edu.cn; wchengx@mail.sysu.edu.cn

approaches have achieved satisfactory results. Since the structural deformations start at the electrode/electrolyte interface, stabilization of the surface structure plays a critical role in preventing Mn dissolution. Lu et al. recently proposed a cation substitution method that involved the conversion of Mn-based Prussian blue to Fe-substituted Prussian blue, which reduced Mn dissolution and promoted highly reversible potassium-storage properties[24]. However, to the best of our knowledge, a high-energy and stable ASIB with an MnPB-based positive electrode has not been reported.

In this work, Fe-substituted Prussian blue $Na_{1.58}Fe_{0.07}Mn_{0.97}$ $Fe(CN)_6 \cdot 2.65H_2O$ is first employed as a cathode material for ASIBs. We present an unconventional in situ remediation strategy by introducing a cation-trap agent $Na_4Fe(CN)_6$ into a concentrated electrolyte (17.6 m $NaClO_4$, m represents molality) to rapidly capture soluble $Mn^{2+}$. Given the lattice expansion that occurs upon $Na^+$ extraction, the anion $Fe(CN)_6^{4-}$ is introduced to coordinate with the dislocating or dislocated Mn to repair the Mn vacancies in situ and enhance surface chemical stability. As a result, the structural integrity of the cathode material can be well preserved, and the ASIBs deliver initial discharge capacities of 157 mAh $g_{cathode}^{-1}$ and 125 mAh $g_{cathode}^{-1}$ at 0.5 A $g^{-1}$ and 10 A $g^{-1}$, respectively.

## Results

### Design of the concentrated aqueous electrolyte solution

The research idea for the electrolyte design came from our experiments in preparing Fe-substituted MnPB (denoted as NaFeMnF). It is generally accepted that the rapid reaction rate achieved using conventional coprecipitation methods usually causes a large amount of $Fe(CN)_6$ vacancies, sodium deficiency, and low crystallinity, all of which lead to poor electrochemical performance[25,26]. Standard strategies for decreasing vacancies involve reducing the reaction rate by introducing a chelating agent and raising the alkaline ion concentration. Recently, Huang et al. explored a postsynthetic method to repair the vacancy defects of iron hexacyanoferrate in a highly concentrated $Na_4Fe(CN)_6$ solution[27]. In this work, we employed an $HNO_3$, $NaSO_4$, and $Na_4Fe(CN)_6$ mixed solution to modify presynthesized MnPB (denoted as NaMnF). Upon exchanging Mn with Fe, homogenous Fe-doped NaMnF with high crystallinity was fabricated. During the experiment, the dissolution of NaMnF in acid solution could be inhibited effectively in $Na^+$-rich or $Na_4Fe(CN)_6$ solutions, particularly in mixtures (for details, please see Supplementary Figs. 1–4, Supplementary Tables 1–2, and Supplementary Note 1). Inspired by this, we proposed an unusual electrolyte consisting of highly concentrated 17.6 m $NaClO_4$ and 0.33 m $Na_4Fe(CN)_6$ to increase the energy content and improve the stability of ASIBs.

### Material characterizations

The X-ray diffraction (XRD) patterns of the two materials (NaMnF and NaFeMnF) shown in Fig. 1a, b demonstrate the well-crystallized diffraction peaks of the cubic and monoclinic structures, respectively. In comparison with NaMnF, the diffraction peaks of NaFeMnF shift to the left, suggesting altered unit cell parameters. Some sharp peaks at ~23.5°, 37.7°, 48.5°, and 54.9° appear in the XRD pattern of NaFeMnF, which could be ascribed to the increased sodium content and phase transition from the cubic to monoclinic phase[28]. The Rietveld refined XRD peaks confirm that the NaFeMnF material is monoclinic with $P2_1/n$ symmetry and that the lattice parameters differ from those of cubic NaMnF (Supplementary Tables 2–4). The insets in Fig. 1a, b present the typical crystal structures of the two materials. The thermogravimetric analysis (TGA) in Fig. 1c shows that the water content of NaFeMnF is 13.98 wt%, lower than that of 18.53 wt% in NaMnF. By combining the inductively coupled plasma-atomic emission spectroscopy (ICP–AES) (Supplementary Table 5) and TGA results, the chemical formulas of the two materials are calculated to be $Na_{1.39}Mn_{1.04}Fe(CN)_6 \cdot 3.65H_2O$ and $Na_{1.58}Fe_{0.07}Mn_{0.97}Fe(CN)_6 \cdot 2.65H_2O$, respectively.

Raman spectroscopy measurements were carried out to measure structural changes (Fig. 1d). Generally, the stretching vibrations of the C≡N group occur in the range of 2050–2200 $cm^{-1}$, corresponding to coordination with transition-metal ions in different valence states[29,30]. For the NaFeMnF material, a new peak centered at 2110 $cm^{-1}$ appears, but a broad peak centered at ~2063 $cm^{-1}$ and a shoulder peak at 2080 $cm^{-1}$ nearly disappear compared with the NaMnF material. The new peak can be assigned to the stretching vibration of $Fe^{2+}$–C≡N–$Fe^{3+}$, suggesting the formation of $Fe^{2+}$–C≡N–$Fe^{3+}$ bonds after modification[31]. The broad peak centered at ~2063 $cm^{-1}$ and the shoulder peak at 2080 $cm^{-1}$ can be attributed to the free C≡N⁻ in Fe–C≡N that does not coordinate with Mn species[32]. The presence of free C≡N⁻ implies the formation of Mn vacancies, which can be formed when a strong chelating agent is used[33]. The decreased intensities of this broad peak and vanishing shoulder peak confirm the decreased defect concentration of Mn vacancies, matching well with the Fe-substituted Mn vacancies to form $Fe^{2+}$–C≡N–$Fe^{3+}$. In addition, both peaks at 2088 and 2131 $cm^{-1}$, belonging to the stretching vibrations of $Fe^{2+}$–C≡N–$Mn^{2+}$ and $Fe^{2+}$–C≡N–$Mn^{3+}$, respectively, are divided into two peaks[31,34]. This may be because of the changed local crystallographic symmetry caused by Fe introduction[35]. It is evident that both the replacement of Mn and the filling of Mn vacancies by Fe effectively diminish crystal effects, which are conducive to accommodating more $Na^+$. For evaluation of electrochemical performance, NaMnF and NaFeMnF were paired with 3,4,9,10-perylenetetracarboxylic diimide (PTCDI) to assemble full cells. As expected, a 50% improvement in specific capacity was achieved by PTCDI‖NaFeMnF (144 mAh $g_{cathode}^{-1}$) in 17.6 m $NaClO_4$ over PTCDI‖NaMnF (96 mAh $g_{cathode}^{-1}$) in 17.6 m $NaClO_4$, as displayed in Fig. 1e.

Figure 1f–m illustrates scanning electron microscopy (SEM) and scanning transmission electron microscopy energy dispersive analysis (STEM-EDS) elemental mapping images of the two materials, respectively. Compared with NaMnF, NaFeMnF has a more regular morphology due to recrystallization during modification, which could indicate better crystallinity in PBA, as previously reported[14]. Despite the improvement in specific capacity, the long-term cycling stability of PTCDI‖NaFeMnF in 17.6 m $NaClO_4$ electrolyte is unsatisfactory (Supplementary Fig. 5).

### Effects of electrolyte engineering

It is known that during the charging process in ASIBs, the Mn–$N_6$ octahedra in MnPB change from stable $Mn^{2+}$ to unstable $Mn^{3+}$, inducing Mn dissolution and Mn vacancy formation on the surface, as seen in Fig. 2a. Continuous Mn dissolution results in structural deformation and rapid capacity decay. Although atom substitution could improve performance, the effects are very limited. Therefore, if the dislocating Mn can be trapped and nucleated deposits can subsequently grow epitaxially on the substrate, Mn dissolution could be mitigated and even prevented. As mentioned above, we used a low concentration of $Na_4Fe(CN)_6$ as an $Mn^{2+}$ trapping agent in a 17.6 m $NaClO_4$ electrolyte. The $Fe(CN)_6^{4-}$ wraps around the surface of each NaFeMnF particle to capture dislocating Mn in situ (Fig. 2b), maintaining structural integrity. In the following discussion, 17.6 m $NaClO_4$ electrolyte is denoted as the blank electrolyte, and 17.6 m $NaClO_4$ + 0.33 m $Na_4Fe(CN)_6$ is denoted as the modified electrolyte. To verify modifications, EDS analyses of the residual content of Mn in the cycled positive electrode (fully discharged electrodes after 100 cycles and 300 cycles in PTCDI‖NaFeMnF in blank electrolyte and modified electrolyte at 0.5 A $g^{-1}$ at 25 °C) were performed. The results in Fig. 2c, Supplementary Fig. 6, and Supplementary Table 6 reveal that the Mn content using the blank electrolyte is reduced with an increased number of cycles. In contrast, in the modified electrolyte, the Mn content is almost unchanged, which proves the suppression of Mn dissolution. Consequently, PTCDI‖

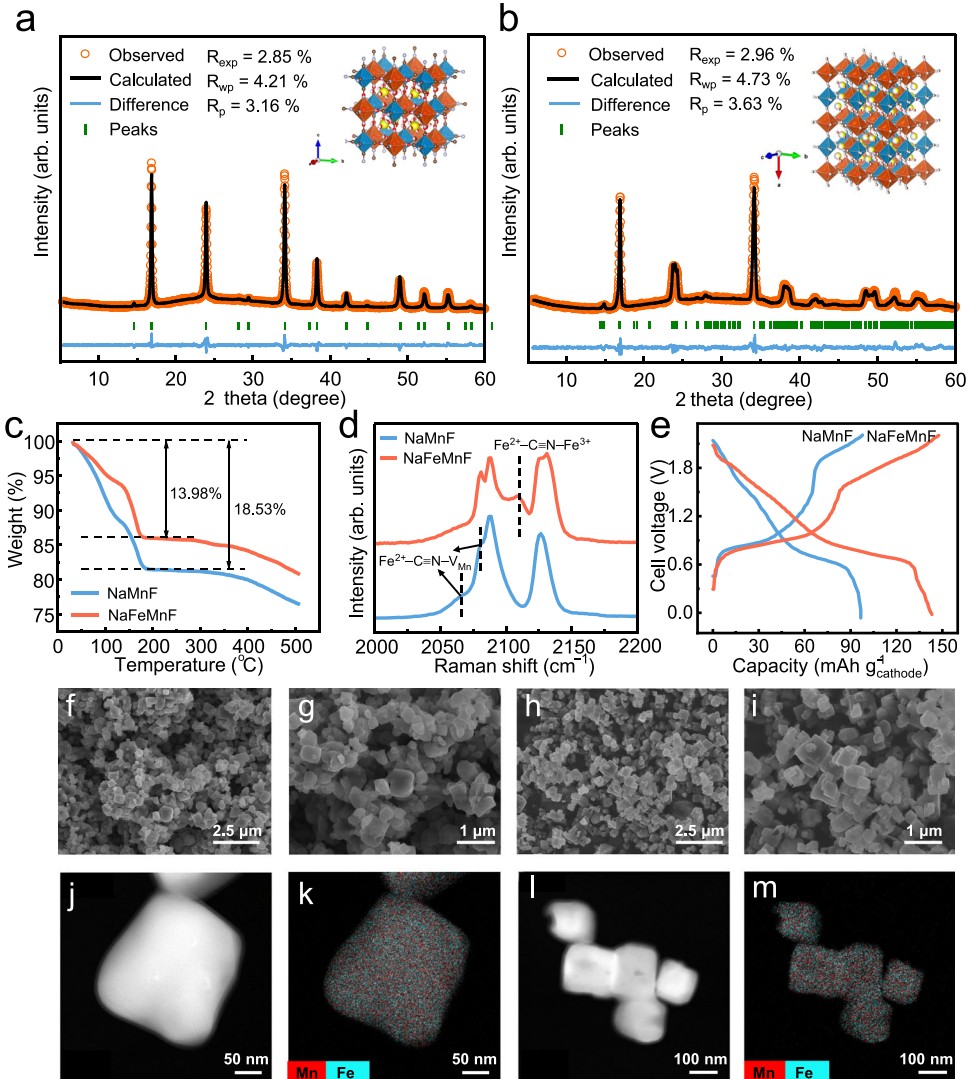

**Fig. 1 | Materials characterization. a** XRD Rietveld refinement results of NaMnF (Mn-based Prussian blue) with an inset of the cell structure. **b** XRD Rietveld refinement results of NaFeMn (Fe-substituted Mn-based Prussian blue) with an inset of the cell structure. In the insets, the orange and blue octahedra represent Mn−N$_6$ and Fe−C$_6$, respectively. The yellow, brown, silvery, and red spheres represent Na, C, N, and O atoms, respectively. **c** TGA results of NaMnF and NaFeMnF powders. **d** Raman spectra of NaMnF and NaFeMnF powders. **e** The tenth charge/discharge profiles of PTCDI‖NaFeMnF and PTCDI‖NaMnF in 17.6 m NaClO$_4$ at 2 A g$^{-1}$ at 25 °C. **f, g** SEM images of NaMnF powder. **h, i** SEM images of NaFeMnF powder. **j, k** STEM-mapping images of NaMnF powder. **l, m** STEM-mapping images of NaFeMnF powder.

NaFeMnF in the blank electrolyte and modified electrolyte displays different cycling stability, as shown in Fig. 2d. An initial discharge capacity of 142 mAh g$_{cathode}^{-1}$ and capacity retention of 54.6% after 600 cycles are obtained at 0.5 A g$^{-1}$ in the blank electrolyte, both of which are significantly improved to 157 mAh g$_{cathode}^{-1}$ and 95.6% with the aid of the Mn$^{2+}$ trapping agent. We compare the Na-ion storage capability of the material in this work with that from previous studies based on five parameters (Fig. 2e and Supplementary Table 7): cycle number, Coulombic efficiency, specific capacity, average discharge voltage, and specific energy, which demonstrate a well-rounded electrochemical performance for our ASIB.

**Evaluation and characterization of the electrolyte**

"Water-in-salt" (WIS) electrolytes are considered a feasible solution for anchoring free-state water and broadening the electrochemical stability window (ESW) of aqueous electrolytes[36]. To probe the ESW of electrolytes, linear sweep voltammetry measurements were carried out. As shown in Fig. 3a, the ESW is extended when the concentration

of the NaClO$_4$-based electrolyte solution increases from 8.8 to 17.6 m, which arises from breakage of the water network by ClO$_4^-$, which has a strong tendency to participate in Na$^+$-solvation[37]. The modified electrolyte provides a similar ESW over 3.6 V relative to the blank electrolyte, indicating that the additional Na$_4$Fe(CN)$_6$ has almost no influence on the ESW. In this case, cyclic voltammetry (CV) tests on coin cells consisting of activated carbon‖NaFeMnF and PTCDI‖activated carbon were used to evaluate the cathodic and anodic limits, respectively. With the widened ESW, charge–discharge of high-potential NaFeMnF and low-potential PTCDI can be fully carried out in the modified electrolyte. The CV curves of the PTCDI‖NaFeMnF full cell shown in Fig. 3b illustrate several reversible redox peaks at 0.31/ 0.36 V, 0.46/0.74 V, 1.82/2.02 V, which are related to the oxidation/ reduction of Fe(CN)$_6^{4-}$/Fe(CN)$_6^{3-}$, Fe$^{2+}$/Fe$^{3+}$, and Mn$^{2+}$/Mn$^{3+}$ couples, respectively. Another pair of redox peaks at -1.46/1.48 V can be explained by the relative redox energy shift and overlap of partial Mn$^{2+}$/Mn$^{3+}$ couples and Fe$^{2+}$/Fe$^{3+}$ couples. This gradually occurs due to poor electronic conductivity with Na$^+$ extraction/insertion[13,38]. The broad redox peaks at -1 V might be caused by Na$^+$ insertion/extraction

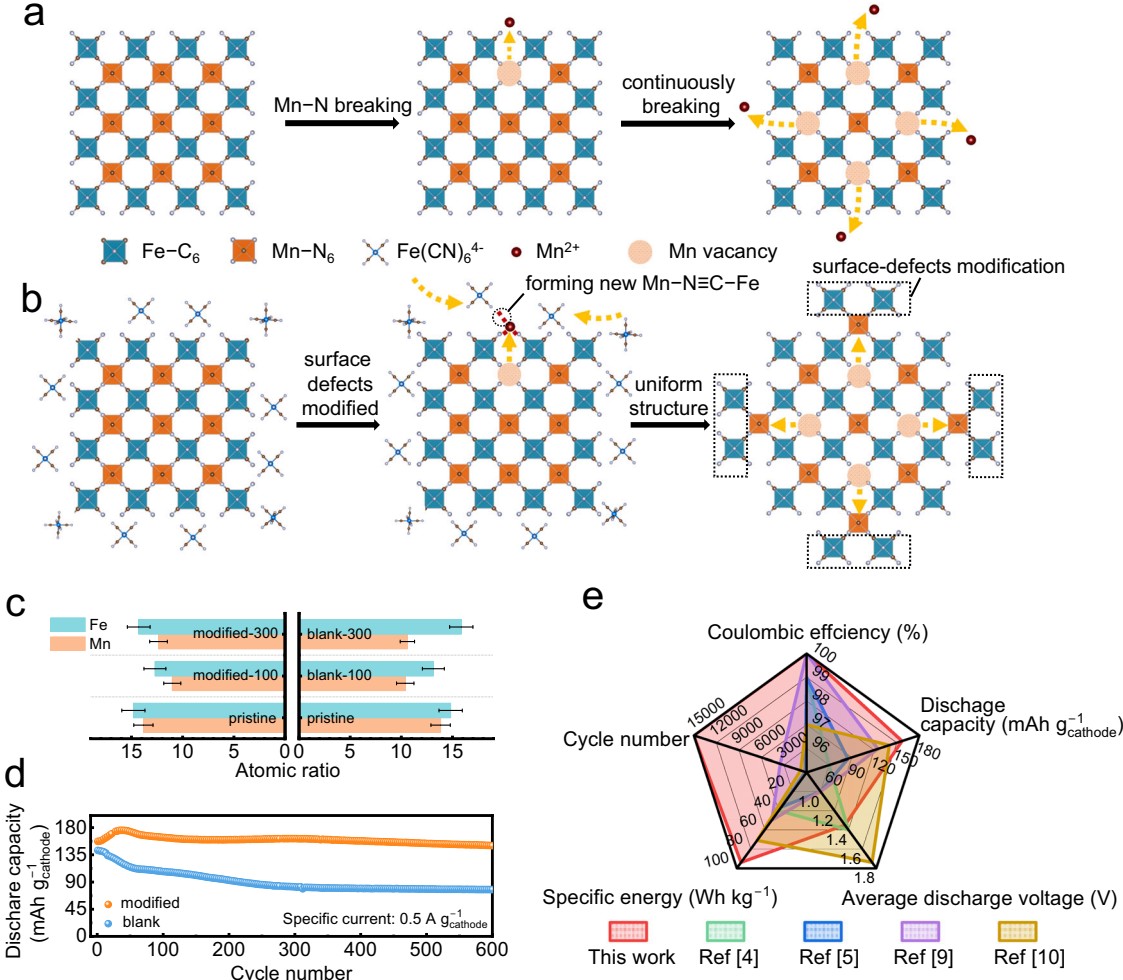

**Fig. 2 | Schematic illustration of the electrolyte engineering strategy.**
**a** Schematic illustration of the Mn dissolution process. **b** Schematic illustration of the cation-trapping process. **c** The atomic ratio of Mn and Fe in the cycled cathode in the blank (17.6 m NaClO₄) and modified (17.6 m NaClO₄ + 0.33 m Na₄Fe(CN)₆) electrolytes as determined by ex situ EDS. The error bars indicate a 68.4% confidence interval. **d** Comparison of the cycling performance of PTCDI∥NaFeMnF in the blank and modified electrolytes at 25 °C. **e** Comparison of the performance of some previously reported ASIBs with those in this work.

from other sites in the structure[39,40]. The results clearly prove that the modified electrolyte can provide a sufficiently wide ESW for high-voltage ASIBs.

Nuclear magnetic resonance (NMR), Fourier transform infrared spectroscopy (FTIR), and Raman spectroscopy were employed to clarify the interaction between water molecules and ions. As shown in Fig. 3c, the $^1$H NMR spectra exhibit a redshift with increasing NaClO₄ concentration and a shift to the lowest value in both the 17.6 m NaClO₄ electrolyte and 17.6 m NaClO₄ + 0.33 m Na₄Fe(CN)₆ electrolyte. This confirms that a stronger interaction between ions and water molecules occurs at higher concentrations of NaClO₄. Figure 3d shows that the infrared adsorption corresponding to the strong H-bond (3000–3500 cm$^{-1}$) almost disappears and that of the weak H-bond (3500–3600 cm$^{-1}$) becomes dominant for the highly concentrated electrolytes, evidencing the stabilized H₂O in the modified electrolyte. The bonding effects between water and salts were evaluated by Raman spectroscopy. As displayed in Fig. 3e, f and Supplementary Fig. 7, both the characteristic Raman bands of the ClO₄$^-$ stretching vibration mode (934 cm$^{-1}$) and C≡N stretching vibration modes (2060 and 2090 cm$^{-1}$) show a distinct blueshift, which is attributed to the transformation of free ClO₄$^-$/Fe(CN)₆$^{4-}$ anions to solvent-separated ion pairs and contact ion pairs. In other words, almost all the ClO₄$^-$/Fe(CN)₆$^{4-}$ anions are coordinated with Na$^+$ or water molecules in highly concentrated solutions, consistent with previous

studies[36,41]. Interestingly, there are no Raman shifts involving the Na₄Fe(CN)₆ additive in comparison with the blank electrolyte, implying that the addition of Na₄Fe(CN)₆ does not change the solvation structure of NaClO₄·H₂O.

It is usually assumed that the rapid fading of Prussian blue analogs in aqueous electrolytes is induced by the metal-ion dissolution of the materials in water[42]. The suppressed activity of free water in WIS electrolytes efficiently reduces the content of the insoluble-to-soluble PBA transformation, which is advantageous for enhancing AIB cycling stability. However, the WIS strategy often fails and is not suitable for Mn-based PBAs. Previous studies on the Na-ion storage mechanism of NaMnF in (non)aqueous batteries imply that NaMnF undergoes a three-phase transition from the monoclinic to the cubic and ultimately to the tetragonal phases[28]. Essentially, Mn dissolution, which is caused by the large volume changes that occur during phase transitions to generate Mn vacancies, cannot be addressed through the individual WIS strategy. Given that Mn dissolution and the migration of Fe(CN)₆$^{4-}$ anions towards positive electrodes cooccurred during the charging process, our in situ cation-trapping method is expected to reconnect dislocating or even dislocated Mn to the surface. The strengthened surface stability is validated by the improved reversibility of redox peaks belonging to the Mn$^{2+}$/Mn$^{3+}$ redox couple in the CV curves of the PTCDI∥NaFeMnF coin cell in the modified electrolyte (Fig. 3b) compared to those of the PTCDI∥NaMnF and PTCDI∥NaFeMnF coin cells in

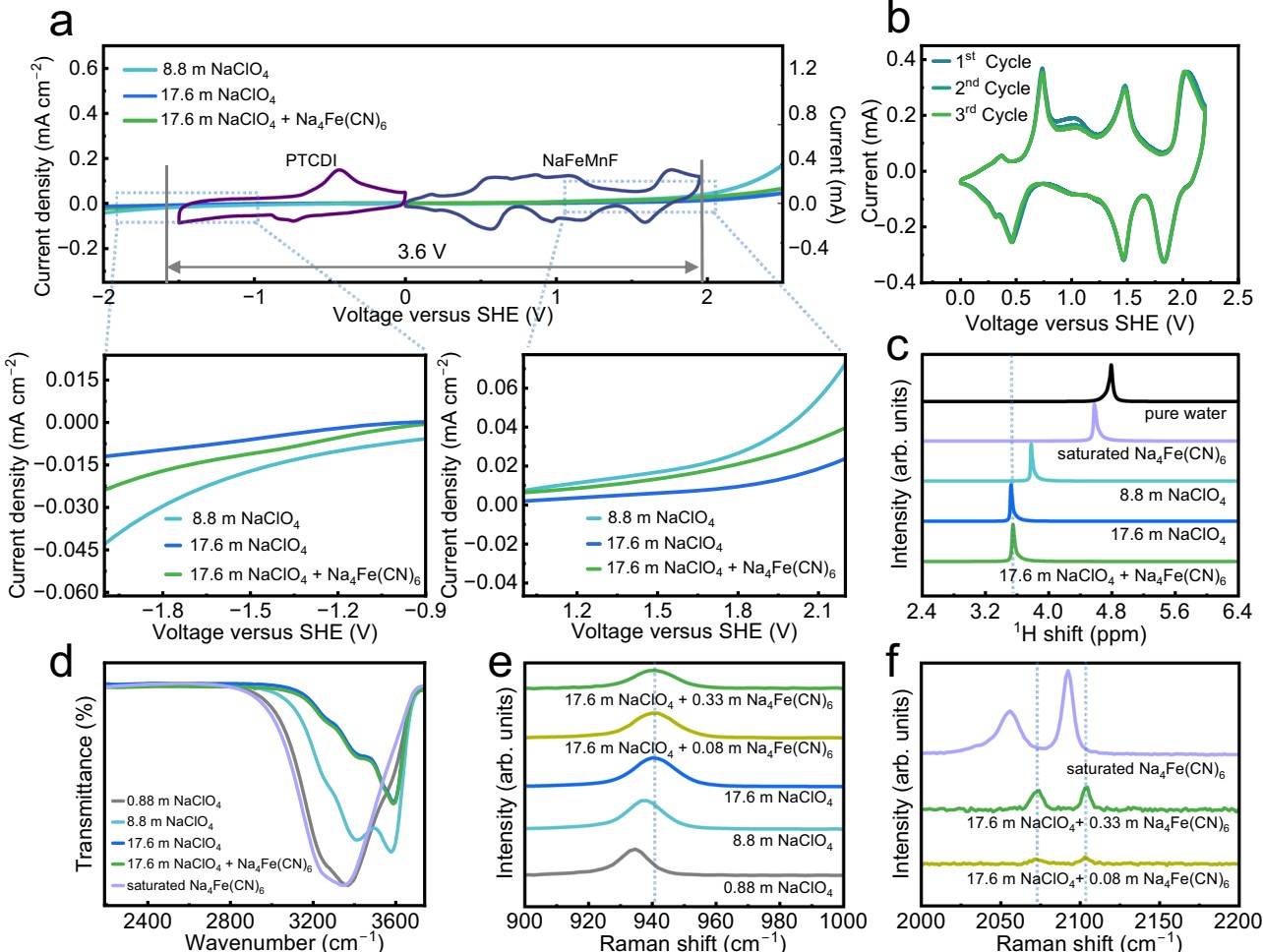

**Fig. 3 | Electrolyte characterization. a** Overall electrochemical stability window of different electrolytes on Ti electrodes obtained by linear sweep voltammetry, with cyclic voltammograms of PTCDI‖activated carbon and activated carbon‖NaFeMnF in the modified electrolyte overlaid. **b** Cyclic voltammograms of PTCDI‖NaFeMnF in the modified electrolyte. **c** The $^1H$ NMR chemical shift in different solutions.

**d** FTIR spectra of the OH stretching mode for different solutions. **e** Raman spectra of $ClO_4^-$ stretching mode for different solutions. **f** Raman spectra of the C≡N stretching mode for different solutions. All characterizations were carried out at 25 °C.

the blank electrolyte (for details, please see Supplementary Fig. 8 and Supplementary Note 2). To further verify the cation-trapping mechanism, studies on the morphological and structural evolutions of NaFeMnF-based positive electrodes (tested in the ASIB coin cell configuration) based on various characterizations were investigated.

### Investigation of the charge storage mechanism and NaFeMnF structural evolution

Figure 4a illustrates a contour plot of in situ XRD measurements of NaFeMnF-based electrode (in PTCDI‖NaFeMnF configuration using the modified electrolyte solution) during the charge–discharge processes (for the XRD patterns, please see Supplementary Fig. 9). Upon charging from 0 to 1.8 V, the peaks at 16.7°, 23.7°, and 33.9° slightly shift to lower angles, corresponding to a phase change from monoclinic to cubic. With further charging from 1.8 to 2.2 V, the peaks at ~16.5° and 23.5° shift to higher angles, while the peaks at ~33.9° and 34.9° gradually vanish, indicating a phase change from cubic to tetragonal. In the subsequent discharge process, all the peaks return to those of the pristine monoclinic phase, suggesting a reversible phase transition (Fig. 4b)[33]. The charge–discharge curves collected at the 1st and 300th cycles of the PTCDI‖NaFeMnF coin cell in different electrolytes are shown in Supplementary Fig. 10. When the blank electrolyte is used, the cell delivers a high discharge plateau (~1.8 V) in the first cycle,

corresponding to the reduction of $Mn^{3+}$ to $Mn^{2+}$, which then disappears gradually with extended cycling. The low-potential region of 0.5–0.9 V, representing the redox couple of $Fe^{2+}/Fe^{3+}$, increases to a higher potential. These results could be attributed to Mn dissolution from the cathode and increased electrochemical polarization. In contrast, the charge/discharge plateaus are well maintained in the modified electrolyte.

To understand how the introduction of $Na_4Fe(CN)_6$ enhances structural integrity, we systematically characterized the cycled NaFeMnF in the fully discharged state (the 100th and 300th cycles) by ex situ SEM, EDS, electron paramagnetic resonance (EPR) spectra, Raman spectra, STEM along with EDS mappings and energy-loss spectroscopy (EELS) measurements. The SEM images (Supplementary Fig. 11) of NaFeMnF-based positive electrodes cycled in the blank and modified electrolytes after 100 cycles demonstrate different particle morphologies. The noticeable surface deformation makes it difficult to distinguish an individual NaFeMnF particle in the former, but the latter electrode shows a particle morphology that is consistent with that of the uncycled electrode. The atomic ratio of Mn to Fe determined by EDS (Supplementary Fig. 6 and Supplementary Table 6) indicates Mn loss in the electrode. The values decrease from 0.92 to 0.67 and slightly to 0.86 in the two electrolytes, respectively, revealing that the addition of $Na_4Fe(CN)_6$ to the electrolyte is favorable for anchoring Mn

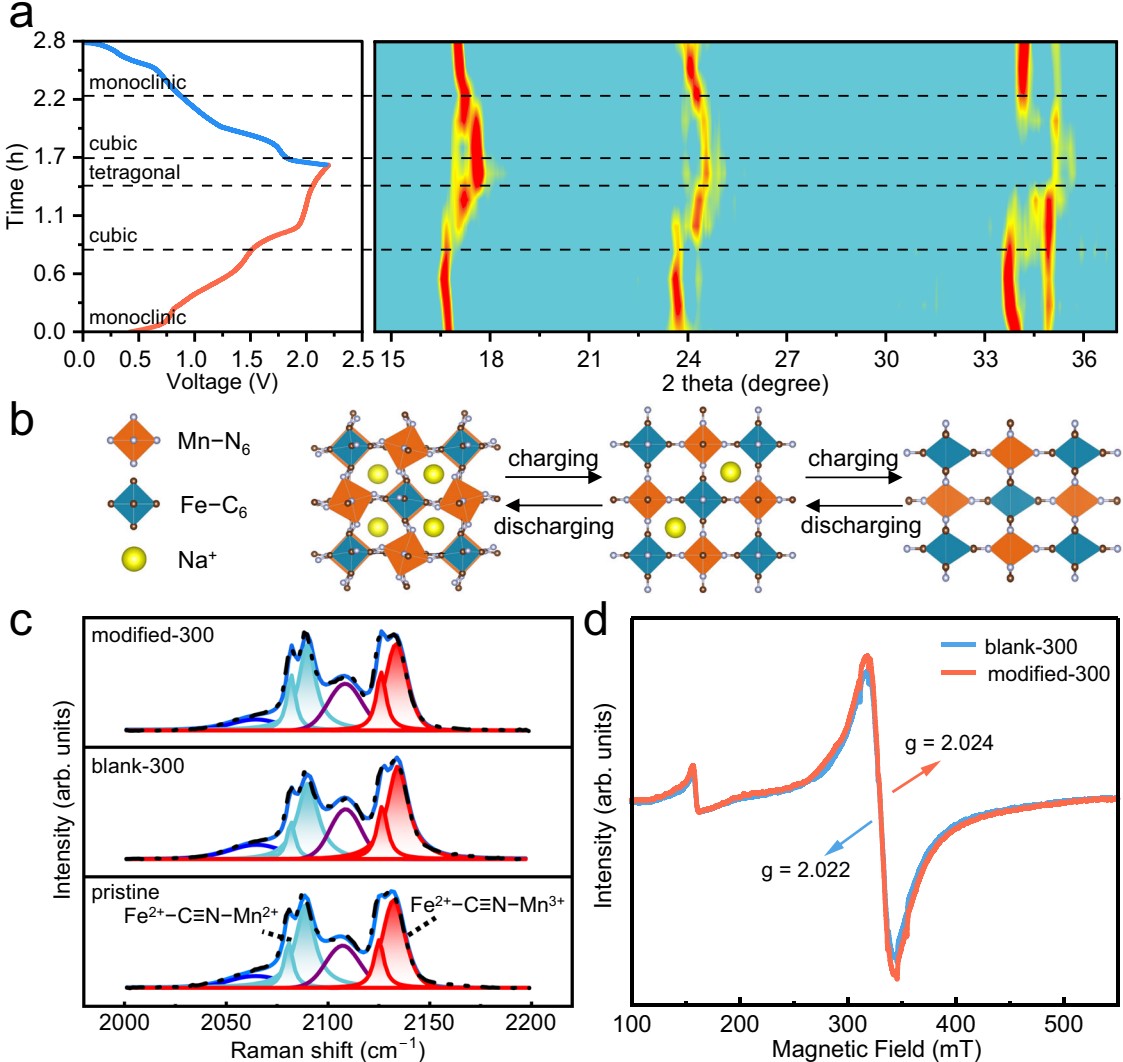

**Fig. 4 | Na-ion storage mechanism in the NaFeMnF-based positive electrode. a** In situ XRD and corresponding charge-discharge profiles of PTCDI||NaFeMnF in the modified electrolyte at 50 mA g⁻¹ and 25 °C. **b** Unit cell structural changes of NaFeMnF during three phase transitions. **c** Ex situ Raman measurement of fully discharged cathodes in the blank electrolyte and modified electrolyte. The blue peaks correspond to $Fe^{2+}–C≡N–Mn^{2+}$ vibrational modes, and the red peaks correspond to $Fe^{2+}–C≡N–Mn^{3+}$ vibrational modes. **d** Ex situ EPR measurements of fully discharged cathodes in the blank electrolyte and modified electrolyte. PTCDI|| NaFeMnF was subjected to 300 cycles at 0.5 A g⁻¹ at 25 °C before collection and ex situ Raman and ex situ EPR measurements of the positive electrodes.

to the electrode. Raman spectroscopy can help us to directly understand the local bonding configuration of the residual Mn in the discharged electrodes, as shown in Fig. 4c and Supplementary Fig. 12. Two pairs of split peaks (2080/2088 cm⁻¹ and 2124/2132 cm⁻¹) can be assigned to the stretching vibrations of $Fe^{2+}–C≡N–Mn^{2+}$ (color in blue) and $Fe^{2+}–C≡N–Mn^{3++}$ (color in red), respectively. The intensity ratios of the two stretching modes in the Raman spectra are 0.72:1 and 0.96:1 for the NaFeMnF cathodes in the blank and modified electrolytes, respectively, indicating that an irreversible phase transition in NaFeMnF occurs in the blank electrolyte, which induces the formation of an electrochemically inactive material. Interestingly, the crystal structures of the cycled electrodes in both electrolytes remain stable (Supplementary Fig. 13). Therefore, it is reasonable to assume that structural deformation leads to amorphization of the surface structure, which blocks Na⁺ insertion. Figure 4d shows the EPR spectra of the fully discharged positive electrodes after 300 cycles (for the EPR spectra of electrodes after 100 cycles, please see Supplementary Fig. 14). It has been reported that $Mn^{2+}$ shows a strong signal in EPR, while $Mn^{3+}$ is not detectable at an ambient temperature of 25 °C[43–45]. Therefore, the stronger EPR signal could be attributed to the relatively

higher content of $Mn^{2+}$ in the electrode cycled in the modified electrolyte. To further prove this view, the g factor, reflecting spin-orbit interactions with the matrix environment[45], was calculated. Compared with the electrode cycled in the blank electrolyte, the electrode cycled in the modified system shows a higher g factor, which results from more $Mn^{2+}$ with high unpaired electron density[46]. This further proves that our cation-trapping strategy is effective in anchoring $Mn^{2+}$ to the electrode, which agrees with the above discussions.

Figure 5a, b presents the ex situ SEM images of cycled electrodes in different electrolytes after 300 cycles, which show that the cubic shape is well maintained in the modified electrolyte. For electrode cycling in the blank electrolyte, the particles adhere to each other so that the grain framework outline is not clear. The EDS mapping image of such individual particles in Fig. 5c shows obvious Mn aggregation. Instead, Mn in the electrode is evenly distributed across the whole particle (Fig. 5d), demonstrating the effectively suppressed Mn dissolution. To probe compositional variations, an STEM-EELS line scan was performed on the cross-section of particles collected from the cycled electrodes. As shown in Fig. 5e, f, we randomly chose three positions (insets in

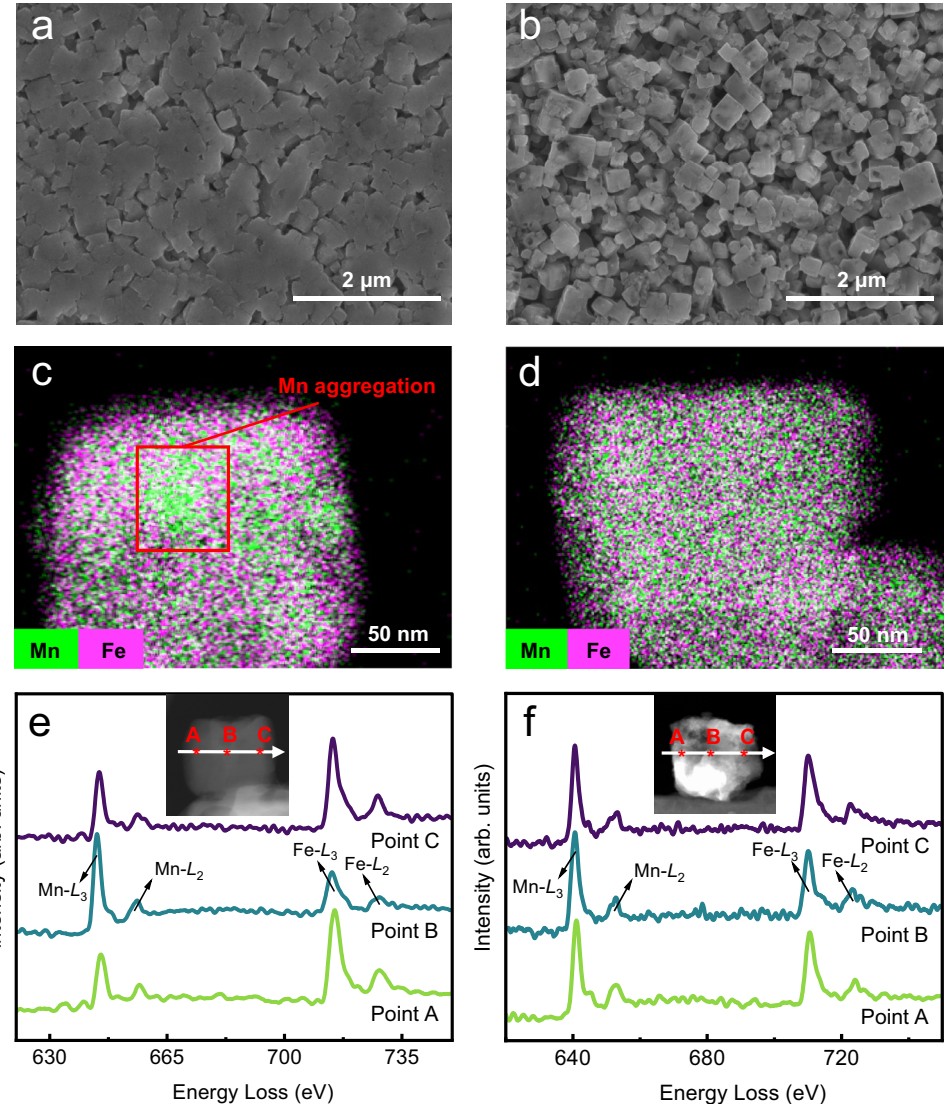

**Fig. 5 | Ex situ physicochemical characterizations of cycled positive electrodes.**
**a**, **b** SEM images of fully discharged electrodes in the blank electrolyte and the modified electrolyte. **c**, **d** STEM-mapping images of fully discharged electrodes in the blank electrolyte and the modified electrolyte. **e**, **f** EELS line scan of fully discharged electrodes in the blank electrolyte and the modified electrolyte. The blank system is on the left, and the modified system is on the right. PTCDI‖NaFeMnF was subjected to 300 cycles at 0.5 A g⁻¹ at 25 °C before the collection and characterization of the positive electrodes.

Fig. 5e, f) to study the valence states of Mn and Fe (for more information, please see Supplementary Fig. 15). A sharp increase in the Mn $L_3/L_2$ intensity ratio from 4.02 (point A) to 5.18 (point B) is detected in the blank system. The ratio variation implies the inhomogeneous valence states of Mn in an individual particle and unreduced high-valence Mn, which are in good agreement with the Raman spectra shown in Fig. 4c. This might be related to the amorphous products generated from structural collapse. In contrast, for the modified system, the Mn $L_3/L_2$ intensity ratio is almost unchanged, demonstrating the homogeneity of Mn valence states in the particle. In addition, there is a noticeable variation in the intensity ratio of Mn to Fe from point A to point B in the blank electrolyte. This implies Mn dissolution in the exterior, which is consistent with the EDS mapping results.

In summary, with the aid of the $Na_4Fe(CN)_6$ additive, the $Mn^{2+}/Mn^{3+}$ redox couple remains highly reversible after long-term cycling, which is confirmed by Raman, EPR, and STEM analysis. Considering the low solubility ($8.0 \times 10^{-13}$ mol L⁻¹ at 25 °C) of $Mn_2Fe(CN)_6$, it is assumed that coordination between $Mn^{2+}$ and $Fe(CN)_6^{4-}$ occurs during cycling to trap Mn.

## Electrochemical performance

We evaluated the PTCDI‖NaFeMnF full cell with a negative-to-positive mass ratio of ~1.25 in the blank and modified electrolytes. As Fig. 6a depicts, the full cell operated in the blank electrolyte delivers an initial discharge capacity of 141 mAh $g_{cathode}^{-1}$ at a current of 0.5 A g⁻¹, which rapidly attenuates to 108 and 82 mAh g⁻¹ in the 100th and 300th cycles, respectively. The charge/discharge profiles change during cycling. With the modified electrolyte, the full cell shows stable charge/discharge voltage plateaus and delivers a high specific capacity of 157 $g_{cathode}^{-1}$ with negligible decay (Fig. 6b). Interestingly, the cation-trapping strategy enables long-term cell cycling stability. As illustrated in Fig. 6c, the full cell exhibits capacity retention of 73.4% after 15,000 cycles at 2 A $g_{cathode}^{-1}$, where the specific capacity remains high at 47.5 mAh g⁻¹ (calculated based on the mass of active materials of both electrodes) with Coulombic efficiency close to 100%, much better than those in the blank system (Supplementary Fig. 16). Furthermore, a good rate capability at various specific currents (0.5–10 A g⁻¹) is also achieved. The full cells deliver capacities (based on the mass of active materials of both electrodes) of 78, 76, 67, 59, and 53 mAh g⁻¹ at various specific currents from 0.5 to 1, 2, 5, and 10 A g⁻¹, respectively (Fig. 6d

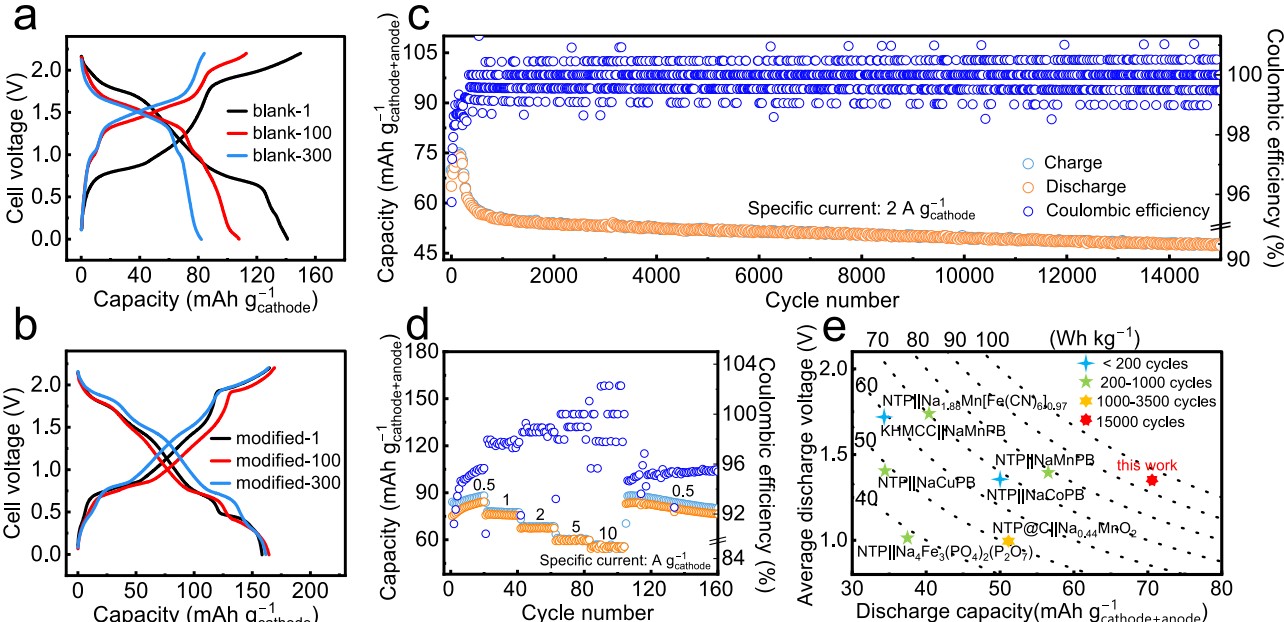

**Fig. 6 | PTCDI‖NaFeMnF cell testing.** The 1st, 100th, and 300th charge–discharge profiles in the blank electrolyte at 0.5 A g⁻¹ at 25 °C (**a**) and modified electrolyte (**b**). **c**, **d** Cycling performance at 2 A g⁻¹ and rate performance (from 0.5 to 1, 2, 5, and 10 A g⁻¹) in the modified electrolyte at 25 °C. **e** Comparison of the capacity, average voltage, specific energy, and cycling stability of this cell in work with those of other previously reported ASIBs. NaMnPB, NaCuPB, NaCoPB, NTP, and KHMCC represent $Na_2MnFe(CN)_6$, $Na_2CuFe(CN)_6$, $Na_2CoFe(CN)_6$, $NaTi_2(PO_4)_3$, and $K_{0.01}Mn[Cr(CN)_6]_{0.72}\cdot2.01H_2O$, respectively.

and Supplementary Fig. 17). When the specific current returns to 0.5 A g⁻¹, the cell is able to deliver a capacity of more than 76 mAh g⁻¹. The favorable electrochemical performance could be attributed to suppressed Mn dissolution. Since NaFeMnF retains surface chemical stability, the charge-transfer resistances in different cycles are consistently lower than those in the blank electrolyte; meanwhile, the diffusion coefficient of Na in the material is higher than that in the blank electrolyte (for details, please see Supplementary Figs. 18–19 and Supplementary Note 3). Figure 6e and Supplementary Table 8 summarize the electrochemical performance of the cell in this work and other reported ASIBs[5,9,10,22,23,47,48]. The PTCDI‖NaFeMnF coin cell demonstrated a calculated specific energy of 94 Wh kg⁻¹ at 0.5 A g⁻¹ (specific energy calculated based on the mass of active materials of both electrodes).

## Discussion

In summary, we developed an aqueous "water-in-salt" electrolyte by introducing a cation-trapping agent ($Na_4Fe(CN)_6$) that hinders Mn dissolution in Mn-based Prussian blue analogs. The nonflammable and highly concentrated "water-in-salt" electrolyte widened the electrochemical stability window over 3.0 V. $Na_4Fe(CN)_6$ did not change the local coordination among $Na^+$, $ClO_4^-$, and $H_2O$; instead, it not only contributes to additional capacity based on the redox reaction of $Fe(CN)_6^{4-}/Fe(CN)_6^{3-}$ but also plays a crucial role in repairing surface defects in situ to prevent Mn loss and structural deformation. We engaged in careful and thorough investigations on the crystal/surface structure and morphological changes in the electrode materials in blank and modified electrolytes. The sharp contrast in the structural integrity and elemental uniformity provides visible evidence to support our speculations. In contrast to conventional doping-induced modifications and electrolyte engineering methods, this cation-trapping strategy is a promising countermeasure for alleviating Jahn–Teller distortion. The resulting aqueous sodium-ion full cell, which is made up of an iron-substituted manganese hexacyanoferrate cathode and an organic anode, yields specific energy of 94 Wh kg⁻¹ at 0.5 A g⁻¹ and 73% discharge capacity retention after 15,000 cycles at 2 A g⁻¹.

## Methods
### Materials
$Na_4Fe(CN)_6$ (analytical reagent, 99.7%) and $MnAc_2\cdot4H_2O$ (analytical reagent, 99.7%) were purchased from Shanghai Aladdin. Tetrasodium ethylenediaminetetraacetate dihydrate (EDTA tetrasodium, 99%) and 3,4,9,10-perylenetetracarboxylic diimide (95%) were purchased from Macklin. $Na_2SO_4$ (99%) and $NaClO_4$ (ACS reagent, >98%) were purchased from Sigma–Aldrich. All the chemicals and reagents were used without further purification.

### Preparation of NaMnF
NaMnF, an Mn-based Prussian blue, was synthesized by a coprecipitation method with the assistance of a chelating agent. First, 0.76 g EDTA tetrasodium and 0.98 g $MnAc_2\cdot4H_2O$ were dissolved in 50 mL deionized water, labeled solution A. $Na_4Fe(CN)_6$ (1.23 g) was dissolved in another 50 mL of deionized water, labeled solution B. Then, solution A was added dropwise into solution B with magnetic stirring for 12 minutes. After that, the mixture was aged for 4 h. Finally, the precipitate was washed with deionized water three times, and dried at 60 °C in air overnight.

### Preparation of NaFeMnF
NaFeMnF, a modified Mn-based Prussian blue by $HNO_3$, $Na_4Fe(CN)_6$, and $Na_2SO_4$, was prepared by the modification of NaMnF. First, 0.2 g NaMnF, 0.12 g $Na_4Fe(CN)_6$, and 10 g $Na_2SO_4$ were mixed in 40 mL deionized water and placed in a water bath at 90 °C with constant stirring. Then, 0.2 mL $HNO_3$ at 1 mol L⁻¹ was added. Afterward, the suspension was further stirred at 90 °C for 4 h. Finally, the precipitate was washed with deionized water three times and dried at 60 °C under air overnight.

### Preparation of other Prussian blue analogs
A series of Prussian blue analogs were fabricated in this work, labeled PB-S1, PB-S2, PB-S3, PB-S4, and PB-S5. PB-S1, an Mn-based Prussian blue, was obtained by the same route as NaMnF preparation. PB-S2, a modified Mn-based Prussian blue by $HNO_3$, was synthesized as follows: a dispersion of 0.2 g NaMnF in 40 mL

deionized water was placed in a water bath at 90 °C with constant stirring. Then, 0.2 mL HNO₃ at 1 mol L⁻¹ was added. Afterward, the dispersion was further stirred at 90 °C for 4 h. Finally, the precipitate was washed with deionized water three times and dried at 60 °C in air overnight. PB-S3, a modified Mn-based Prussian blue by HNO₃ and Na₄Fe(CN)₆, was synthesized by the same route as PB-S2 except 0.12 g Na₄Fe(CN)₆ was added to the above dispersion before the addition of acid. PB-S4, a modified Mn-based Prussian blue by HNO₃ and Na₂SO₄, was synthesized by the same route as PB-S2 except 10 g Na₂SO₄ was added to the above dispersion before the addition of acid. PB-S5, a modified Mn-based Prussian blue by HNO₃, Na₄Fe(CN)₆, and Na₂SO₄, was obtained by the same route as NaFeMnF.

### Preparation of electrolyte
The blank electrolyte was prepared by dissolving 24.5 g NaClO₄ in 10 mL of deionized water. The modified electrolyte was prepared by dissolving 24.5 g NaClO₄ and 1 g Na₄Fe(CN)₆ in 10 mL of deionized water. Laboratory ultrapure water system produced the above-deionized water with a resistivity of 18.2 MΩ·cm.

### Materials characterization
All the cycled cells for ex situ measurements were disassembled in air to collect the electrodes. The collected electrodes were thoroughly washed with deionized water for at least 10 min before further investigation. Storage and transport of the electrode samples were performed under an air atmosphere. The XRD data of the materials and electrodes were measured on a Rigaku diffractometer (Cu Kα radiation, λ = 0.154 nm). The TGA data of the cathode materials were collected under N₂ flow at a ramp-up rate of 5 °C min⁻¹ from 30–500 °C by a thermogravimetric analyser (TG-209, Netzsch). The Raman data of the electrolytes and electrodes were collected by an inVia Qontor Raman microscope with a 532 nm laser. The SEM images of the electrodes were observed using a Regulus 8230 at 10 kV, and EDS was obtained at 10 kV. ¹H NMR spectra of electrolytes were collected on a 400 MHz Bruker NMR spectrometer at ambient temperature. The FTIR spectra of the electrolytes were collected by a Fourier transform infrared spectrometer (NICOLET 6700). EPR spectra of the cycled electrodes were collected by a Bruker A300-10/12 at room temperature. High-angle annular dark-field images (HAADF), STEM-mapping images, and EELS line scans of the electrodes were obtained by TEM (FEI Tecnai G2 F30, 300 kV) with an accelerating voltage of 300 kV using a lacey support film. ICP–AES tests of materials were performed by (Optima8300).

### Electrochemical measurements
The NaFeMnF material and PTCDI were used as the cathode and anode in this work. Electrode slurries were prepared by mixing in a mass ratio of 70 wt% active materials, 20 wt% carbon black (100 nm, 99.5%, purchased from Macklin), 10 wt% polyvinylidene fluoride (PVDF, 99.5%, purchased from Solvay) and an appropriate amount of N-methyl pyrrolidone (NMP, 99.9%, purchased from Shanghai Aladdin). The mixing to prepare the slurry was carried out by a human operator using mortar and pestle. The slurries were coated on carbon paper (0.05 mm, 99%, purchased from XFNANO, China) and dried at 80 °C under air overnight. Then, the electrodes were pressed under a pressure of 10 MPa. Each carbon paper was loaded with 1.0–2.0 mg cm⁻² of active material for a 2032-type coin-cell. The mass ratio of the cathode/anode was -1/1.25. Both blank electrolytes (17.6 m NaClO₄) and modified electrolytes (0.08–0.33 m Na₄Fe(CN)₆ per 17.6 m NaClO₄) were used for testing. Separators (GF/A, with a thickness of 0.26 mm and diameter of 150 mm) were purchased from Whatman. For optimized performance, the coin cells were preactivated for 5 cycles at 0.5 A g⁻¹ in a climate-controlled chamber with the temperature set to 25 °C. The

preactivation cycles are not considered in the calculations of capacity retention, specific energy, etc.

All electrochemical measurements were carried out in a climate-controlled chamber with the temperature set to 25 °C. Ti foil (0.05 mm, 99.6%) used was purchased from Alfa Aesar. Activated carbon (200 mesh) used was purchased from Macklin. Linear sweep voltammetry was carried out in Ti||Ti in various electrolytes at a scan rate of 0.5 mV s⁻¹. Cyclic voltammetry was carried out in PTCDI||activated carbon, activated carbon||NaFeMnF, PTCDI||NaMnF, and PTCDI||NaFeMnF in the modified electrolyte at a scan rate of 0.5 mV s⁻¹. Electrochemical impedance spectroscopy (EIS) data were obtained over the frequency range of 100 kHz to 0.1 Hz at a voltage amplitude of 5 mV at the open circuit voltage of the cells (10 data points per tenfold change in frequency). The abovementioned tests were conducted on a CHI 600c electrochemical workstation and a DH7003 electrochemical workstation. Galvanostatic discharge–charge measurements, rate performance measurements, and the galvanostatic intermittent titration technique (GITT) were carried out within 0–2.2 V using a Neware battery system (Neware Technology Ltd.). GITT was conducted at a specific current of 40 mA g⁻¹, in which the cell was alternately charged for 30 min followed by 10 min of rest and then discharged in the same way. The specific current was based on the active material mass of the positive electrodes unless otherwise stated. The specific capacity was based on the active material mass of both electrodes unless otherwise stated. The specific energy was based on the active material mass of both electrodes. At least five cells have been tested for an individual electrochemical experiment.

### Reporting summary
Further information on research design is available in the Nature Portfolio Reporting Summary linked to this article.

## Data availability
The data generated in this study have been deposited on Figshare (https://doi.org/10.6084/m9.figshare.23283695).

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

## Acknowledgements

C.W. acknowledges the support from the National Natural Science Foundation of China (U1801255 and 91963210). G.Y. acknowledges the support from the National Natural Science Foundation of China (51872340) and the Natural Science Foundation of Guangdong Province, China (2021A1515010143).

## Author contributions
G.Y. and C.W. conceived and supervised the project; Z.L. and G.Y. designed the experiments; Z.L. performed the experiments with the help of F.T.; Z.L., F.T., G.Y., and C.W. discussed the results; Z.L., G.Y., and C.W. wrote the initial paper which was approved by all the authors.

## Competing interests
The authors declare no competing interests.
