## [Peer Review File · Nature Communications]

REVIEWER COMMENTS

Reviewer #1 (Remarks to the Author):

This is a very interesting and important work regarding the stabilization of Mn-based Prussian blue analogue material to be used as cathode for aqueous Na-ion batteries. The study is well-conducted, the results are convincing, the battery was demonstrated with high stability and good performance. A structure/property relationship is presented and several in situ analysis have been carried out to explain some results. I believe that this paper is adequate to be published in Nature Communications, after authors consider some specific points:

1. some recent papers on electrodes based on PBA for Na-ion batteries are missing, and authors should cite it: 1) *Electrochimica Acta* 422 (2022) 140548; 2) *Composites Part B: Engineering* (2022), 246, 110241; *ACS Sustainable Chem. Eng.* 2022, 10, 13277–13287
2. TGA analysis: authors said "The thermogravimetric analysis (TGA) analysis in Figure 1c indicates that NaFeMnF contains a lower content of water than that of NaMnF", and it's all. Why? What are the consequences?
3. Raman spectra: authors affirm that "The broad peak centered around 2063 cm^{-1} and the shoulder peak at 2080 cm^{-1} can be attributed to the characteristic of free $[\text{C}\equiv\text{N}]^-$ ". What is free $[\text{C}\equiv\text{N}]^-$? Are you considering adsorbed species that don't are effectively bonded to the structure, or the ones linked to the Mn species at the surface proposed in this work? Is it possible to differentiate them? What is the origin of this free $[\text{C}\equiv\text{N}]^-$ and the implications on the properties of the materials?
4. The proposed chemical formulas of the two materials were obtained based on two different techniques: one of them is a surface-based technique (EDS) and the other one a bulk technique (ICP). How have authors mixed the information from these different techniques to achieve the formulas presented in the manuscript? What's the reliability of this approach?
5. the caption for Fig 1 doesn't match with Fig. 1. (m) in Fig 1 is missing. It was not possible to see who for the microscopy images. This way, some discussion was missed, as for example, when authors say: "Compared with NaMnF, NaFeMnF has more regular morphology, moreover, its particle has more uniformed size and better crystallinity". Anyway, it's dangerous attribute crystallinity of a material based on a SEM image".
6. What are the data that can prove the proposal sketched in Fig. 2b, which is the basis of this work? The absence of Mn release in the NaFeMnF material means that something happened and avoided the Mn release, but cannot prove that the proposed mechanism sketched in Fig 2b is happens.
7. The phrase "Another pair of redox peaks at around 1.46/1.48 V can be explained by the Mn energy level splitting, which is ascribed to the poor conductivity of cathode when Na^+ extracted" looks speculative and confuse. Authors should make efforts to clarify this point.
8. the relationship between the intensity of the EPR signal and the ration between $\text{Mn}^{2+}/\text{Mn}^{3+}$ is not consistent. Authors said that " Mn^{2+} species (electronic configuration $3d^5$) is expected to show a higher

EPR signal than that of Mn³⁺". It should be true if the amount of both is the same. A stronger EPR signal shouldn't come from higher amount of Mn³⁺ compared to Mn²⁺?

9. related to the Mn aggregation seen by EDS in Figure 5b: is it reproducible? Have authors seen other regions like that, in other images?

Reviewer #2 (Remarks to the Author):

Battery system using Fe-doped NaMnF cathode for aqueous Na-ion batteries have been investigated. The ion-type cell for NaFeMn/PTCDI has high energy density and good cyclability, and the discussion is interesting. However, reviewers have questions following;

I believe the concentration of a saturated solution of NaClO₄ is about 17 mol/kg. How can it be prepared at a concentration of 20 mol/kg? If possible, please tell me the weight of the salt and the solvent. Also, is the battery measured temperature room temperature or 25 oC?

What is the initial pH of the electrolyte?

The average voltage is stated as 1.5 V (Fig. 2 e). Looking at the charge/discharge curve, I think the average voltage is about 1.0 V. How did you determine the average voltage?

In Fig. 6 (a), (b), the discharge capacity around 1.8 V with blank was larger than that of modified electrolyte. Why is this?

Isn't the content that must be claimed most in this paper about the electrolyte with Na₄Fe(CN)₆ added? Electrolytes are not mentioned in the abstract. I think it is necessary.

For ion type cell, the capacity decreased until 200-300 cycles. Is this an anode effect?

In addition, the current density of 2 A/g is high rate. What will be the cyclability at lower rates? Will self-discharge occur when stored in a charged state?

REVIEWER COMMENTS

Reviewer #1 (Remarks to the Author):

This is a very interesting and important work regarding the stabilization of Mn-based Prussian blue analogue material to be used as cathode for aqueous Na-ion batteries. The study is well-conducted, the results are convincing, the battery was demonstrated with high stability and good performance. A structure/property relationship is presented and several in situ analysis have been carried out to explain some results. I believe that this paper is adequate to be published in Nature Communications, after authors consider some specific points:

Reply: Dear Reviewer, thanks very much for your positive feedback, we appreciate for your insightful comments and constructive suggestions on our manuscript entitled “*Ultrastable High-Energy Aqueous Sodium-Ion Batteries Enabled by Suppressing Mn Dissolution in Prussian Blue Cathode*” (Research paper, Manuscript ID: NCOMMS-22-43851). All comments are valuable and helpful for improving the quality of our paper. We have carefully studied these comments and made corresponding revisions, in hope of addressing your concerns and meeting the high standards of *Nature Communications*. The point-by-point replies for comments were listed as follows, and relevant revisions have been highlighted in the revised manuscript.

1. some recent papers on electrodes based on PBA for Na-ion batteries are missing, and authors should cite it: 1) *Electrochimica Acta* 422 (2022) 140548; 2) *Composites Part B: Engineering* (2022), 246, 110241; *ACS Sustainable Chem. Eng.* 2022, 10, 13277–13287

Answer: Thank you very much for the valuable suggestion. We have read through these papers and agreed that these papers could enrich our content. As recommended, we have cited these papers in the manuscript. Please see the Ref. 15, 16, 17.

2. TGA analysis: authors said! The thermogravimetric analysis (TGA) analysis in Figure 1c indicates that NaFeMnF contains a lower content of water than that of NaMnF”, and it’s all. Why? What are the consequences?

Answer: Thank you very much for pointing out this problem. The original description is indeed not specific enough. TGA is a common characterization method for determining the water content of Prussian blue analogues^{1,2}. The mass loss below 300 °C is normally assigned to the water molecules in the lattice. Seen in Fig. 1c, NaFeMnF shows 13.98% weight loss below 300 °C which is lower than 18.53% of NaMnF, indicating less content of water. Notably, there has been reported that water molecules would occupy guest cation storage sites and coordinating with transition metal causing Fe(CN)₆ vacancies^{3,4}. Herein, NaFeMnF with less water content contains more sodium storage sites, which is supported by the results of ICP-AES (Supplementary Table 5) and XRD (Fig. 1b). Meanwhile, distinct improvement in capacity is achieved for NaFeMnF, displayed in Fig. 1e. The sentence has been updated as follow: The thermogravimetric analysis (TGA) in Fig. 1c shows that the water content of NaFeMnF is 13.98 wt%, lower than 18.53 wt% of NaMnF.

References

- (1) Wang, W. *et al.* Reversible structural evolution of sodium-rich rhombohedral Prussian blue for sodium-ion batteries. *Nat. Commun.* **11**, 980 (2020).
- (2) Song, J. *et al.* Removal of interstitial H₂O in hexacyanometallates for a superior cathode of a sodium-ion battery. *J. Am. Chem. Soc.* **137**, 2658–2664 (2015).
- (3) You, Y., Wu, X. L., Yin, Y. X. & Guo, Y. G. High-quality Prussian blue crystals as superior cathode materials for room-temperature sodium-ion batteries. *Energy Environ. Sci.* **7**, 1643–1647 (2014).
- (4) Peng, J. *et al.* Prussian Blue Analogues for Sodium-Ion Batteries: Past, Present, and Future. *Adv. Mater.* **34**, 2108384 (2022).

3. Raman spectra: authors affirm that “The broad peak centered around 2063 cm⁻¹ and the shoulder peak at 2080 cm⁻¹ can be attributed to the characteristic of free [C≡N]⁻”. What is free [C≡N]⁻? Are you considering adsorbed species that don’t are effectively bonded to the structure, or the ones linked to the Mn species at the surface proposed in this work? Is it possible to differentiate them? What is the origin of this free [C≡N]⁻ and the implications on the properties of the materials?

Answer: Thanks for your careful reading and bringing this meaningful point to our attention. We agree that the original description is indeed not specific enough. We consider that the above-mentioned peaks actually come from $\text{C}\equiv\text{N}^-$ in $\text{Fe}-\text{C}\equiv\text{N}$ that do not coordinate with Mn species, which implies the formation of Mn vacancies. Similar result in Raman spectra had been reported previously and explained by the usage of strong chelating agent^{1,2}. By contrast, Prussian blue with high crystallinity shows sharp peaks rather than broad peaks or shoulder peaks³. Therefore, we believe that both the broad peak and the shoulder peak should come from the Mn vacancy. As you mentioned, the difference of peak positions between them should be ascribed to different coordination environments of the surface and bulk phases. To be specific, the broad peak around 2063 cm^{-1} should come from the surface $\text{C}\equiv\text{N}^-$ which has lower stability than the one in the bulk phase. Consistent with the above discussion, the cycled electrode with more Mn vacancies on the surface also shows a stronger peak around 2063 cm^{-1} in Fig. 4c. In addition, the shoulder peak around 2080 cm^{-1} should come from the $\text{C}\equiv\text{N}^-$ in the bulk. Therefore, the NaFeMnF with Mn vacancies filled by Fe shows no shoulder peaks around 2080 cm^{-1} in Fig. 1d.

Furthermore, Mn vacancies of inappropriate concentration would lead to the collapse of $\text{Fe}-\text{C}\equiv\text{N}-\text{Mn}$ bridge and the reduction of redox-active sites. This is one of the reasons for the poor electrochemical performance of NaMnF. In contrast, NaFeMnF with Mn vacancies filled by Fe shows better electrochemical performance, displayed in Supplementary Fig. 5. In addition, electrodes with stable Mn content during cycling deliver superior electrochemical performance, displayed in Fig. 6c. For clear expression, the original sentence has been updated as follow: The broad peak centered around 2063 cm^{-1} and the shoulder peak at 2080 cm^{-1} should be attributed to the surface $\text{C}\equiv\text{N}^-$ and bulk $\text{C}\equiv\text{N}^-$ that do not coordinate with Mn species. The presence of free $\text{C}\equiv\text{N}^-$ implies the formation of Mn vacancies, which will be easily formed when using strong chelating agent.

References

- (1) Moretti, G. & Gervais, C. Raman spectroscopy of the photosensitive pigment Prussian blue. *J. Raman Spectrosc.* **49**, 1198–1204 (2018).

- (2) Shang, Y. *et al.* Unconventional Mn Vacancies in Mn–Fe Prussian Blue Analogs: Suppressing Jahn-Teller Distortion for Ultrastable Sodium Storage. *Chem* **6**, 1804–1818 (2020).
- (3) Vertelman, E. J. M. *et al.* Light- And temperature-induced electron transfer in single crystals of $\text{RbMn}[\text{Fe}(\text{CN})_6]\cdot\text{H}_2\text{O}$. *Chem. Mat.* **20**, 1236–1238 (2008).

4. The proposed chemical formulas of the two materials were obtained based on two different techniques: one of them is a surface-based technique (EDS) and the other one a bulk technique (ICP). How have authors mixed the information from these different techniques to achieve the formulas presented in the manuscript? What's the reliability of this approach?

Answer: We apologize for the careless mistake we made. As a matter of fact, we did not mix the information of both EDS and ICP to achieve the chemical formulas. Instead, we determined the atomic ratio by ICP. In the process of writing, we mistakenly wrote TGA as EDS. As suggested, we believe that EDS, as a surface analysis technique, cannot accurately provide atomic information of the bulk phase, so it is inappropriate to determine the chemical formula of the sample by EDS results. In fact, our original idea was to use ICP to determine the atomic ratio of metal elements, and use TGA to determine the water content in the sample, and combine the two techniques to obtain the chemical formula of samples. Thanks again for your reminder, we have revised the sentence in the manuscript.

5. The caption for Fig 1 doesn't match with Fig. 1. (m) in Fig 1 is missing. It was not possible to see who is who for the microscopy images. This way, some discussion was missed, as for example, when authors say: "Compared with NaMnF , NaFeMnF has more regular morphology, moreover, its particle has more uniformed size and better crystallinity". Anyway, it's dangerous attribute crystallinity of a material based on a SEM image".

Answer: We apologize for the careless mistake in the caption. We have carefully revised each caption and letter marks for Fig. 1, and replaced the original caption with

the corrected one. In addition, we appreciate for your reminder about the description of SEM images. As suggested, it is not acceptable to determine the sample crystallinity merely depending on SEM images. However, in the case of Prussian blue and its analogues, highly crystallized products are usually cubic-shaped¹. Therefore, it could be meaningful to remain the relevant part. Anyway, we suppose there might be some misleading to readers, so the sentence has been updated as follow: Compared with NaMnF, NaFeMnF has more regular morphology thanks to the recrystallization during modification, which could be a symbol of better crystallinity for PBA as previously reported.

Fig. 1 | Electrochemical Material characterizations. f-g SEM images of NaMnF. **h-i** SEM images of NaFeMnF. **j-k** STEM-mapping images of NaMnF. **l-m** STEM-mapping images of NaFeMnF.

Reference

- (1) Peng, J. *et al.* Prussian Blue Analogues for Sodium-Ion Batteries: Past, Present, and Future. *Adv. Mater.* **34**, 2108384 (2022).

6. What are the data that can prove the proposal sketched in Fig. 2b, which is the basis of this work? The absence of Mn release in the NaFeMnF material means that something happened and avoided the Mn release, but cannot prove that the proposed mechanism sketched in Fig 2b is happens.

Answer: Thank you for your crucial comments. We agree this is an important issue and

we propose the surface-repairing mechanism based on characterization from surface to bulk phase. From the SEM images (Fig. 5a, b), the electrodes cycled in the blank electrolyte shows the morphology of particles adhering to each other without distinguishable edges and corners, indicating surface deformation relating to Mn dissolution on the surface. This is very different with the pristine electrode and the electrode cycled in the modified electrolyte with $\text{Na}_4\text{Fe}(\text{CN})_6$ added. On the other hand, the electrodes cycled in the blank and modified electrolytes both show high crystallinity in XRD patterns (Supplementary Fig. 13), demonstrating intact bulk structure of both electrodes. Therefore, we suppose there must be something occurring with the aid of $\text{Na}_4\text{Fe}(\text{CN})_6$ to prevent the surface deformation.

We have carefully considered the above results and made efforts to reveal the mechanism. Firstly, for the fully discharged in the modified electrolyte after 300 cycles, the EDS results (Supplementary Fig. 6 and Supplementary Table 6) shows stable Mn content on the surface compared to the pristine electrode. Furthermore, the STEM-mapping results (Fig. 5d) reveals uniform Mn distribution from the surface to the bulk. These results indicate that the $\text{Na}_4\text{Fe}(\text{CN})_6$ additive could help lock Mn on the surface. Secondly, for the fully discharged in the modified electrolyte after 300 cycles, the intensity ratio of Mn^{2+} and Mn^{3+} in the Raman spectra (Fig. 4c) is almost unchanged compared with the pristine electrode. In addition, EELS (Fig. 5f) shows an almost unchanged Mn L_3/L_2 intensity ratio of a particle from the exterior to the interior, demonstrating the homogeneity of Mn valence states in the particle. These results demonstrate that surface Mn^{2+} were retained in the electrode thanks to interaction with $\text{Na}_4\text{Fe}(\text{CN})_6$. Given to no impurity in the XRD pattern (Supplementary Fig. 13) of the cycled electrode and the low K_{sp} (Solubility Product Constants) of $\text{Mn}_2\text{Fe}(\text{CN})_6$, we believe the $\text{Na}_4\text{Fe}(\text{CN})_6$ is able to capture the soluble Mn^{2+} and reduce Mn dissolution by coordination with Mn. To further prove this view, more advanced *in situ* characterization at atomic resolution should be included. We are sure it would be an interesting process and constructive to our work. However, under the conditions of the limited equipment, we are sorry for not being able to carry out these characterizations.

7. The phrase “Another pair of redox peaks at around 1.46/1.48 V can be explained by the Mn energy level splitting, which is ascribed to the poor conductivity of cathode when Na⁺ extracted” looks speculative and confuse. Authors should make efforts to clarify this point.

Answer: Thanks for your careful reading and reminding us this point. We found this comment very crucial and valuable. The original statement is not precise enough indeed. We consider that this pair of redox peaks come from the potential shifts of Mn²⁺/Mn³⁺ couple and Fe²⁺/Fe³⁺ couple, which are resulted from the poor conductivity as the Na⁺ extraction/insertion proceeds. As displayed in Fig. 3b and Fig. 6a, b, the pair of redox peaks around 1.46/1.48 V in the CV diagram corresponds to the mid-voltage plateau in the charge-discharge curves. Although the mid-voltage plateau is subtle in the first cycle, it become more dominated as the cycle proceeds. Meanwhile, the high-voltage plateau corresponding to Mn²⁺/Mn³⁺ couple, and the low-voltage plateau corresponding to Fe²⁺/Fe³⁺ couple are becoming shorter. Similar results have been reported previously^{1,2}. Combining the previous research, we consider the appearance of the mid-voltage plateau and the corresponding redox peaks around 1.46/1.48 V are ascribed to the potential shifts of partial Mn²⁺/Mn³⁺ couples and Fe²⁺/Fe³⁺ couples. Intrinsically, it could be explained by the redox energy shift and overlap of the two redox couples caused by poor conductivity gradually with the Na⁺ extraction/insertion¹, illustrated in Figure R1.

For clear expression, the original sentence has been updated as follow: Another pair of redox peaks at around 1.46/1.48 V can be explained by the relative redox energy shift and overlap of partial Mn²⁺/Mn³⁺ couples and Fe²⁺/Fe³⁺ couples. This is caused by poor conductivity gradually with the Na⁺ extraction/insertion.

Figure R1 | Relative redox energies of $\text{Fe}^{2+}/\text{Fe}^{3+}$ and $\text{Mn}^{2+}/\text{Mn}^{3+}$ of the electrode before (left) and after (right) Na^+ extraction/insertion.

References

- (1) Wang, L. *et al.* A superior low-cost cathode for a Na-Ion battery. *Angew. Chem. Int. Ed.* **52**, 1964–1967 (2013).
- (2) Gebert, F. *et al.* Epitaxial Nickel Ferrocyanide Stabilizes Jahn–Teller Distortions of Manganese Ferrocyanide for Sodium-Ion Batteries. *Angew. Chem. Int. Ed.* **60**, 18519–18526 (2021).

8. The relationship between the intensity of the EPR signal and the ration between $\text{Mn}^{2+}/\text{Mn}^{3+}$ is not consistent. Authors said that “ Mn^{2+} species (electronic configuration $3d^5$) is expected to show a higher EPR signal than that of Mn^{3+} ”. It should be true if the amount of both is the same. A stronger EPR signal shouldn’t come from higher amount of Mn^{3+} compared to Mn^{2+} ?

Answer: Thanks for your careful reading and valuable comment. We agree that original statement is not persuasive or rigorous enough considering the amount of the Mn contents in the two samples are not the same. After reading more references, we found that Mn^{2+} would show a strong signal in EPR, while Mn^{3+} is almost silent at ambient temperature^{1,2}. Since our experiment was carried out at room temperature, we believe the stronger EPR signal should be attributed to relative higher content of Mn^{2+} in the electrode cycled in the modified electrolyte². To further prove this view, we have supplemented the measurement of the g factor for electrodes. Note that the g factor could be used to differentiate Mn species since it can reflect spin-orbit interactions with

the matrix environment³. As displayed in Figure R2, the electrode cycled in the modified system shows a higher g factor, implying more Mn²⁺ with high unpaired electron density⁴. For more precise expression, we have deleted the sentence “Mn²⁺ species (electronic configuration 3d⁵) is expected to show a higher EPR signal than that of Mn³⁺(electronic configuration 3d⁴)” and updated the Fig. 4d in the manuscript to discuss g factors.

Figure R2 | EPR spectra of fully discharged electrodes after 300 cycles in the blank electrolytes and modified electrolyte.

References

- (1) Banerjee, A. *et al.* On the oxidation state of manganese ions in li-ion battery electrolyte solutions. *J. Am. Chem. Soc.* **139**, 1738–1741 (2017).
- (2) Kim, S. S. *et al.* Searching for biosignatures using electron paramagnetic resonance (EPR) analysis of manganese oxides. *Astrobiology* **11**, 775–786 (2011).
- (3) Riedel, W. *et al.* Magnetic Properties of Reduced and Reoxidized Mn-Na₂WO₄/SiO₂: A Catalyst for Oxidative Coupling of Methane. *J. Phys. Chem. C* **122**, 22605–22614 (2018).
- (4) Li, H. *et al.* Enhanced charge carrier separation of manganese(II)-doped graphitic carbon nitride: Formation of N-Mn bonds through redox reactions. *J. Mater. Chem. A* **6**, 6238–6243 (2018).

9. Related to the Mn aggregation seen by EDS in Figure 5b: is it reproducible? Have authors seen other regions like that, in other images?

Answer: We appreciate your critical comment. The EDS results of Mn aggregation shown in Fig. 5c can be reproducible indeed. Similar phenomena can also be found in many other particles, as shown in the Figure R3.

Figure R3 | STEM-mapping results of fully discharged electrodes after 300 cycles in the blank electrolyte of other regions. **a** Region I. **b** Region II.

Reviewer #2 (Remarks to the Author):

Battery system using Fe-doped NaMnF cathode for aqueous Na-ion batteries have been investigated. The ion-type cell for NaFeMn/PTCDI has high energy density and good cyclability, and the discussion is interesting. However, reviewers have questions following;

Reply: Dear Reviewer, thanks very much for your positive feedback, we appreciate for your insightful comments and constructive suggestions on our manuscript entitled “*Ultrastable High-Energy Aqueous Sodium-Ion Batteries Enabled by Suppressing Mn Dissolution in Prussian Blue Cathode*” (Research paper, Manuscript ID: NCOMMS-22-43851). All comments are valuable and helpful for improving the quality of our paper. We have carefully studied these comments and made corresponding revisions, in hope of addressing your concerns and meeting the high standards of *Nature Communications*. The point-by-point replies for comments were listed as follows, and relevant revisions have been highlighted in the revised manuscript.

I believe the concentration of a saturated solution of NaClO₄ is about 17 mol/kg. How can it be prepared at a concentration of 20 mol/kg? If possible, please tell me the weight of the salt and the solvent. Also, is the battery measured temperature room temperature or 25 °C?

Answer: Thanks for your kind comment and critical reminder. Through checking on the documentaries, we recognize that the concentration of a saturated solution of NaClO₄ is about 17 mol kg⁻¹. In our previous experiments of preparing electrolytes, the 20 mol kg⁻¹ NaClO₄ solution was obtained by dissolving 0.06 mol NaClO₄ (7.3464 g), purchased from Al** Aes** (CAS: 7601-89-0; Sodium perchlorate, anhydrous, ACS, 98.0-102.0%), in 3 g deionized water. After your reminder, we decided to prepare the solution again according to the above route using the same chemical agent and record the dissolution process in Figure R1. Considering that this is an important issue, we purchased another batch of NaClO₄ from Sigma-Aldrich (Product code: 410241; CAS: 7601-89-0; ≥98.0%). However, we cannot prepare a 20 mol kg⁻¹ solution using the new chemical agent.

To identify whether the solution of NaClO_4 (purchased from Al** Aes**) was 20 mol kg^{-1} , ICP-OES (Inductively Coupled Plasma Optical Emission Spectrometer) test was carried out. To be specific, 22.5 mg of the solid power sample (the chemical agent purchased from Al** Aes**) was dissolved into 50 mL of water, and 2 ml of the solution was taken for test. The ICP-OES results show that the mass ratio of Na in the sample is 16.54% . So, the calculated molar mass of the sample is $138.99 \text{ g mol}^{-1}$, which is larger than $122.44 \text{ g mol}^{-1}$ of anhydrous sodium perchlorate. It can be inferred that the sample is hydrate sodium perchlorate. Therefore, the concentration of the NaClO_4 solution used in the experiment was recalculated as 17.6 mol kg^{-1} . Thanks again for your careful reading and reminder. Relative texts in the manuscript and the Figures have been updated.

In addition, all the battery tests are carried out in the incubator with the temperature set at $25 \text{ }^\circ\text{C}$.

Figure R1 | The NaClO_4 solution preparation process.

What is the initial pH of the electrolyte?

Answer: Thanks for your comment. After dissolution, we used a pH meter to determine the pH value of the above solution as 6.09.

The average voltage is stated as 1.5 V (Fig. 2 e). Looking at the charge/discharge curve, I think the average voltage is about 1.0 V. How did you determine the average voltage?

Answer: Thank you for the comment which we find very crucial and meaningful. After reading more references, we found that we confused the concepts of mid-value voltage with average voltage. Therefore, we recalculated average voltage by dividing the energy density by the mass specific capacity and determined the average voltage of 1.35 V. The energy density is obtained by integrating the discharge curve (Figure R2). The mid-value voltage of 1.44 V that we read from the software is displayed in Figure R3. In addition, Fig. 2e and Fig. 6e have been updated based on the average voltage of 1.35 V.

Figure R2 | Schematic illustration of integration of discharge curve.

Cycle ID	RCap_Chg(m...	RCap_DChg(...	Efficiency(%)	Mid-value Vo...
400	159.631	158.100	99.04	1447.4

Figure R3 | The mid-value voltage that we read from the software.

In Fig. 6 (a), (b), the discharge capacity around 1.8 V with blank was larger than that of modified electrolyte. Why is this?

Answer: Thank you for bringing this meaningful point to our attention. We consider that the smaller discharge capacity around 1.8 V with the modified electrolyte is ascribed to the potential shift of $\text{Mn}^{2+}/\text{Mn}^{3+}$ couple. In the initial cycle, the electrode in

the modified electrolyte shows a shorter voltage plateau around 1.8 V and a slope around 1.5 V (Fig. 6b). Similar results have been reported before and explained by potential shift of $\text{Mn}^{2+}/\text{Mn}^{3+}$ couple, which is caused by the change of conductivity and accompanying change in redox energy of $\text{Mn}^{2+}/\text{Mn}^{3+}$ couple^{1,2}. Notably, the potential shift is even more significant in the blank system after the initial cycle. In the subsequent cycles, a third voltage plateau around 1.5 V appears and gradually becomes prominent in the blank system, along with the shortening and disappearance of the other voltage plateaus (Fig. 6a). It indicates the relative redox energies of $\text{Mn}^{2+}/\text{Mn}^{3+}$ couple and $\text{Fe}^{2+}/\text{Fe}^{3+}$ couple, which are located separately initially, shift gradually and finally overlap¹, as shown in the Figure R4. This leads to poor cycle stability (Fig. 6a). In contrast, although with a shorter voltage plateau around 1.8 V in the initial cycle, the modified system delivers a much better cycle stability and shows charge-discharge curves (Fig. 6b) that almost overlap. It also implies the stability of redox energies of $\text{Mn}^{2+}/\text{Mn}^{3+}$ couple and $\text{Fe}^{2+}/\text{Fe}^{3+}$ couple.

Figure R4 | Relative redox energies of $\text{Fe}^{2+}/\text{Fe}^{3+}$ and $\text{Mn}^{2+}/\text{Mn}^{3+}$ of the electrode before (left) and after (right) Na^+ extraction/insertion.

References

- (1) Wang, L. *et al.* A superior low-cost cathode for a Na-Ion battery. *Angew. Chem. Int. Ed.* **52**, 1964–1967 (2013).
- (2) Chou, S. *et al.* Epitaxial Nickel Ferrocyanide Stabilizes Jahn-Teller Distortions of Manganese Ferrocyanide for Sodium-Ion Batteries. *Angew. Chem. Int. Ed.* **60**, 18519–18526 (2021).

Isn't the content that must be claimed most in this paper about the electrolyte with $\text{Na}_4\text{Fe}(\text{CN})_6$ added? Electrolytes are not mentioned in the abstract. I think it is necessary.

Answer: Thanks for your careful reading of our paper and your important comment. We have added the description of electrolyte additives in the abstract.

For ion type cell, the capacity decreased until 200-300 cycles. Is this an anode effect?

Answer: Thank you for your important reminder. After consulting more references, we found that full cells with PTCDI as the anode and Mn-Fe Prussian blue as the cathode would show a decline in capacity in the initial dozens of cycles¹. The PTCDI anode may affect the battery performance to some certain extent¹. However, we suppose the main reason for the capacity fading is the dissolution of Mn and the resulting polarization. As shown in Figure R5, at the current density of 2 A g^{-1} , the modified system remains 95.2% of the initial capacity after 300 cycles. In contrast, the blank system shows capacity retention of 59.6% after 300 cycles. Similar results can be observed at the current density of 0.5 A g^{-1} , shown in Figure R6. After 300 cycles, the capacity of the modified system shows almost no decay, while the blank system remains only 57% of the initial capacity. Combing the Mn dissolution in the cycle process, supported by Raman spectra (Fig. 4), STEM-EDS mapping and EELS (Fig. 5), it is inferred that Mn dissolution should be the main reason for rapid capacity fading in the blank system.

Figure R5 | The comparison of cycling performance at 2 A g^{-1} of full cells in two electrolytes.

Figure R6 | The comparison of cycling performance at 0.5 A g^{-1} of full cells in two electrolytes.

Reference

- (1) Jiang, L. *et al.* Building aqueous K-ion batteries for energy storage. *Nat. Energy* **4**, 495–503 (2019).

In addition, the current density of 2 A/g is high rate. What will be the cyclability at lower rates?

Answer: Thank you for your reminder. We believe cycle stability at low rate is indeed important. The cycling performance at 0.5 A g^{-1} had been included in Figure R6 and the manuscript (Fig. 2d). It is worth noting that 0.5 A g^{-1} is a relatively low current density used in aqueous batteries, because fast kinetics is one of the advantages of aqueous batteries compared to non-aqueous batteries.

Will self-discharge occur when stored in a charged state?

Answer: Thank you for your comments. We have supplemented the electrochemical tests to investigate the self-discharge behavior. Before the self-discharge study, the batteries went through 10 cycles at 0.2 A g^{-1} for activation. As illustrated in Figure R7, both systems went through severe self-discharge in 48 hours, in which the blank system shows voltage attenuation from 2.2 V to 1.07 V and the modified one from 2.2 V to 1.34 V . In fact, self-discharge has been one of the fatal problems in aqueous battery where severe voltage attenuation will occur within tens to hundreds of hours^{1,2}. This could be ascribed to electrochemical instability and shuttle-reaction between anode and

cathode^{3,4}. Furthermore, voltage attenuation would be even more pronounced in the case of unstable SEI film and high-concentration electrolyte^{4,5}, accounting for severe self-discharge in this study. Fortunately, after severe self-discharge, the modified system shows no reversible capacity loss and keeps good electrochemical stability, as shown in the Figure R8. This demonstrates no irreversible reaction or impedance increase during resting⁶. However, we believe drastic voltage attenuation in this study remains a challenge to be solved. We are aware drastic voltage attenuation in this study, which is one of the most challenging problems faced by the practical application of aqueous battery besides cycle stability and energy density. After finishing this study, we would love to make efforts to study this issue in the future.

Figure R7 | The voltage variations in 48 hours after charging to 2.2 V.

Figure R8 | The cycle stability before and after resting.

References

- (1) Zilberman, I., Sturm, J. & Jossen, A. Reversible self-discharge and calendar aging of 18650 nickel-rich, silicon-graphite lithium-ion cells. *J. Power Sources* **425**, 217–226 (2019).

- (2) Shang, W. *et al.* Optimizing the charging protocol to address the self-discharge issues in rechargeable alkaline Zn-Co batteries. *Appl. Energy* **308**, 118366 (2022).
- (3) Trócoli, R., Morata, A., Erinmwingbovo, C., La Mantia, F. & Tarancón, A. Self-discharge in Li-ion aqueous batteries: A case study on LiMn₂O₄. *Electrochim. Acta* **373**, 137847 (2021).
- (4) Xu, J. & Wang, C. Perspective—Electrolyte Design for Aqueous Batteries: From Ultra-High Concentration to Low Concentration?. *J. Electrochem. Soc.* **169**, 030530 (2022).
- (5) Droguet, L., Grimaud, A., Fontaine, O. & Tarascon, J. M. Water-in-Salt Electrolyte (WiSE) for Aqueous Batteries: A Long Way to Practicality. *Adv. Energy Mater.* **10**, 2002440 (2020).
- (6) Grolleau, S. *et al.* Calendar aging of commercial graphite/LiFePO₄ cell - Predicting capacity fade under time dependent storage conditions. *J. Power Sources* **255**, 450–458 (2014).

REVIEWERS' COMMENTS

Reviewer #1 (Remarks to the Author):

All the questions picked up in my previous report have been satisfactorily answered and the suggestions incorporated into the revised version of the manuscript. So, I consider that it is now ready to be published.

Reviewer #2 (Remarks to the Author):

The reviewers' questions were resolved by the authors' answers.

I think it is acceptable.